# DIMS: Channel-dependent and Seasonal-trend Independent Transformer Using Multi-stage Training for Time Series Forecasting

## Abstract

Due to the limited size of real-world time series data, current Transformer-based time series forecasting algorithms often struggle with overfitting. Common techniques used to mitigate overfitting include channel-independence and seasonal-trend decomposition. However, channel-independent inevitably results in the loss of inter-channel dependencies, and existing seasonal-trend decomposition methods are insufficient in effectively mitigating overfitting. In this study, we propose DIMS, a time series forecasting model that uses multi-stage training to capture inter-channel dependencies while ensuring the independence of seasonal and trend components. The computation of channel dependency is postponed to the later stage, following the channel-independent training, while the seasonal and trend components remain fully independent during the early training phases. This approach enables the model to effectively capture inter-channel dependencies while minimizing overfitting. Experiments show that our model outperforms the state-of-the-art Transformer-based models on several datasets.

## 1 Introduction

Time series forecasting is widely utilized across multiple domains such as transportation, energy, meteorology, retail, and finance. With the rapid development of deep learning, numerous studies have applied deep learning algorithms to multivariate time series forecasting. These studies have leveraged CNNs, RNNs, and various ensemble algorithms, which have achieved excellent results in computer vision and natural language processing (NLP) tasks, for time series forecasting and have seen some success. The Transformer architecture has recently demonstrated outstanding performance in fields such as computer vision and NLP, as seen in models like BERT (Vaswani (2017)), GPT (Achiam et al. (2023)), ViT (Dosovitskiy (2020)), and Swin-Transformer (Liu et al. (2021)). Consequently, many studies are now applying the Transformer architecture to time series forecasting tasks, such as in FEDformer (Zhou et al. (2022)) and Autoformer (Wu et al. (2021)).

In contrast to areas such as computer vision and natural language processing, which benefit from ample datasets, real-world time series forecasting frequently encounters limitations in data availability. Acquiring real-world time series data is inherently time-intensive, often requiring substantial periods to gather sufficient observations. For example, if electricity consumption is recorded hourly, it could take more than a year to compile 10,000 data points. This results in a significant temporal cost for acquiring time series data, and over time, the distribution of the data may change, indicating that merely augmenting the dataset does not guarantee enhancements in predictive performance. Due to these inherent characteristics of time series data, certain counterintuitive phenomena have been observed in time series forecasting experiments. For instance, many well-designed time series forecasting models based on Transformer architectures perform worse than simpler linear models (Zeng et al. (2023)). This is not surprising, as Transformer models typically require large datasets to achieve optimal performance, which real-world time series data often cannot provide, leading to rapid overfitting in Transformer models. Encouragingly, PatchTST (Nie et al. (2022)) has achieved superior results to linear models by using a unique patching method and a channel-independent mechanism. The channel-independent mechanism ignores the relationships between channels (di-

mensions) of the time series data, treating each channel as a homogeneous univariate time series. This mitigates overfitting and improves prediction performance, further highlighting the importance of mitigating overfitting to enhance the accuracy of time series forecasting.

As demonstrated by the "simple" models such as DLinear and PatchTST, which achieve excellent performance, simpler methods and architectures often yield better results in time series forecasting tasks. However, "simple is difficult", as it remains a challenge to simultaneously leverage the powerful correlation-capturing capabilities of Transformers while avoiding the overfitting often induced by time series data. In this study, we propose a channel-**D**ependent seasonal-trend-**I**ndependent model using **M**ulti-**S**tage training (DIMS) for time series forecasting. Our model harnesses the powerful feature extraction capabilities of the Transformer architecture and accounts for the dependencies between time series channels while avoiding premature overfitting.

In summary, our main contributions are as follows:

- We propose a season-trend independent pattern representation and training method that maintains the independence of season and trend components for most of the training process. This method prevents interference between the seasonal and trend components during training, thereby alleviating overfitting.
- The dependencies between time series channels are incorporated into the forecasting model. We propose a method that delays the computation of inter-channel dependencies, integrating this calculation only after the model completes the channel-independent time series dimension operations. This ensures that the capture of time series features is not disturbed, ultimately allowing the integration of the inter-channel dependency module to improve prediction performance rather than prematurely causing overfitting. For datasets that do not meet the assumption of channel independence, we modify the positional encoding of the time series to further enhance prediction performance.
- We conduct experiments on seven widely used public real-world datasets, and the results show that our model outperforms state-of-the-art Transformer-based models.

## 2 RELATED WORK

PatchTST (Nie et al. (2022)) divides time series data into several patches and introduces mechanisms such as channel independence and parameter sharing. Crossformer (Zhang & Yan (2023)) employs a patch division method similar to PatchTST, while also introducing cross-channel attention. Its proposed TSA model structure and attention routing algorithm between channels significantly reduce the computational cost of cross-channel attention, which is crucial for high-dimensional time series data. iTransformer (Liu et al. (2023)) treats each channel of the time series as an individual patch, using feedforward neural networks to extract temporal features and employing a Transformer architecture to capture cross-channel dependencies. In experiments with shorter input sequences, iTransformer demonstrated notable improvements.

It is worth noting that for time series forecasting, especially in long-term forecasting tasks, complex models do not necessarily lead to better prediction accuracy. Models like DLinear, NLinear (Zeng et al. (2023)), and RLinear (**?**) use simple linear layers to capture temporal features and adopt a channel-independent mechanism, which entirely ignores inter-channel relationships. These simple models can achieve prediction accuracy comparable to, or even better than, more complex models.

At the same time, some studies have focused on the differing nature of the seasonal and trend components in time series data. Autoformer (Wu et al. (2021)) decomposes time series data layer by layer through a moving average method to extract periodic components. In its encoder, it applies self-correlation to the seasonal component while discarding the trend component, which is only reintroduced during the final prediction phase in the decoder. FEDformer (Zhou et al. (2022)) also separates the trend component from the time series data. Its key frequency domain enhancement module only processes the seasonal component while ignoring the trend, adding the trend component back only in the final prediction output. The seasonal and trend components can be viewed as the high-frequency and low-frequency parts of a time series, respectively. Consequently, some studies have further transformed time series data from the time domain to the frequency domain for analysis. FiTs (Xu et al. (2023)) uses the Fast Fourier Transform (FFT) to convert sequences into the frequency domain and applies a low-pass filter to extract the core features of the sequence. It then

maps the frequency domain information through a linear network, achieving competitive predictive performance using just 10k parameters. JTFT (Chen et al. (2024)) employs the Discrete Cosine Transform (DCT) to extract frequency domain information and combines it with time-domain information for forecasting.

# 3 METHOD

The problem can be modeled as follows: given an input sequence $x_{L,C}$ of data with dimension $C$ and length $L$, the prediction model generates the output $y_{T,C}$, where $T$ represents the length of the prediction window. $X_{H,C}$ denotes the entire training dataset, with a time length of $H$ and the same dimension $C$.

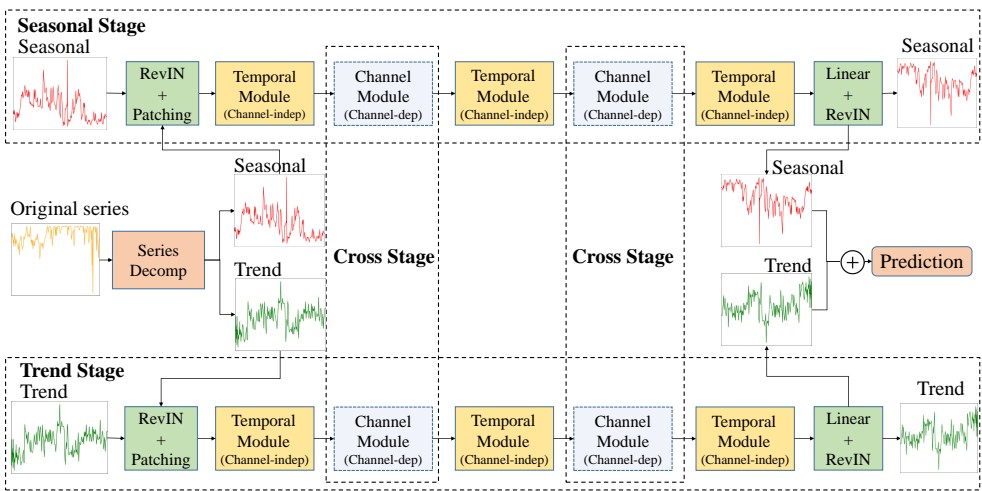

Figure 1: The overall architecture of DIMS. The DIMS model is divided into a seasonal component and a trend component, with each part containing three temporal modules and two channel modules.

## 3.1 SEASONAL AND TREND COMPONENTS

Existing methods for seasonal-trend decomposition typically input the separated seasonal and trend components into two or more modules for synchronized training. While this approach alleviates overfitting, it does not entirely prevent interference between the seasonal and trend components. This is because when producing the final prediction, these methods often sum the predicted seasonal and trend components, causing the gradient of the prediction error to propagate back to both the seasonal and trend prediction modules simultaneously. Since the error reflects the overall time series and does not distinguish between the seasonal and trend components, it interferes with their accurate prediction, leading to faster overfitting.

To avoid the drawbacks of synchronously training the seasonal and trend components, we chose to independently train the prediction models for the seasonal and trend components. As shown in Figure 1, the entire training dataset $X$ is first decomposed into a seasonal component $XS$ and a trend component $XT$. In the seasonal phase, the Transformer structure corresponding to the seasonal component is trained on $XS$. Similarly, in the trend phase, the Transformer structure corresponding to the trend component is trained on $XT$. After completing the training in both the seasonal and trend phases, the two models are combined and jointly trained on the original dataset $X$. In the final training stage, the data is first passed through a moving average module for seasonal and trend decomposition. The decomposed data is then fed into the pre-trained seasonal and trend prediction models, and the predictions from the two models are summed to obtain the final prediction result. It is noted that while we emphasize Seasonal-Trend Independence during the early training phases, the two components are not independent in the final training stage, which allows the model to capture the interdependencies between the seasonal and trend components.

## 3.2 CHANNEL INDEPENDENT OR DEPENDENT

To avoid overfitting, many recent studies have employed channel-independent methods for training and forecasting multivariate time series. Although channel independence has brought significant improvements, handling each dimension of multivariate time series independently while ignoring cross-channel relationships can result in the loss of key information. This makes it unreasonable to rely solely on channel independence for further improving prediction performance. However, introducing cross-channel relationships can indeed lead to overfitting more quickly. For instance, despite Crossformer employing several innovative techniques, it still cannot entirely prevent overfitting, ultimately limiting its predictive performance. Nevertheless, Crossformer provides valuable insights and offers an efficient way to establish cross-channel relationships.

We believe that Crossformer's introduction of cross-channel relationships exacerbates overfitting for two reasons. First, the failure to decompose the seasonal and trend components leads to overfitting, as trend and seasonal components often exhibit different characteristics. Without decomposition, the model struggles to distinguish between these features. In this case, introducing cross-channel attention not only fails to improve prediction accuracy but may also exacerbate overfitting. Second, introducing cross-channel relationships too early interferes with the model's ability to learn temporal features. The most critical characteristic of time series data is the temporal feature within each dimension. Therefore, if cross-channel dependencies are introduced prematurely, it hampers the model's learning of temporal features, negatively impacting the final prediction performance.

To address the first issue, our proposed season-trend independence method effectively resolves the overfitting caused by the lack of decomposition. To tackle the second issue, we delay the calculation of cross-channel dependencies until after the channel-independent computations are completed. The model first performs channel-independent attention calculations, corresponding to the temporal module, focusing exclusively on capturing temporal features within the data. Once the training phase of the channel-independent module is complete, the cross-channel attention module is added to the model, creating a structure similar to a sandwich. A second round of training is then conducted, referred to as the "Cross Stage". This approach allows the model to capture inter-channel dependencies while avoiding interference with the learning of temporal features.

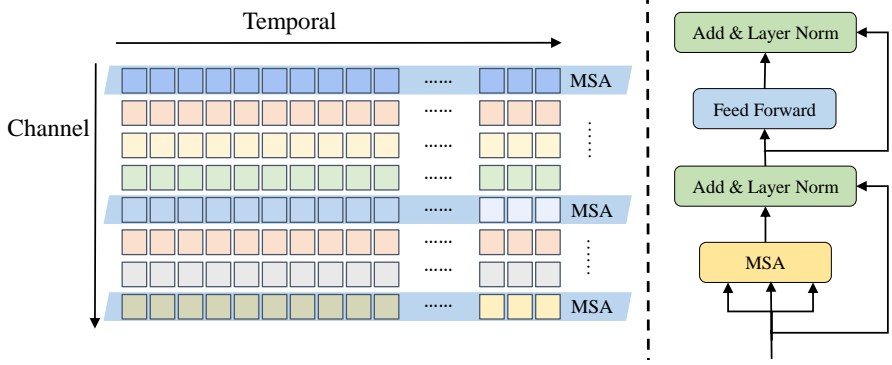

Figure 2: The architecture of the temporal module. The left side shows a schematic diagram of the computational logic applied to the data, while the right side illustrates the module structure.

Therefore, the training in the seasonal stage and trend stage can each be further divided into two stages: the channel-independent "temporal stage" and the channel-dependent "cross stage". The primary operations of the model in the temporal stage involve the temporal module, as shown in Figure2, where the core method is consistent with PatchTST. The input data $x_{L,C}$ is divided into $X_{pC}^d$ through the patching module, followed by multi-head self-attention (MSA) and layer normalization. $E_{dp}$ denotes a set of learnable positional encodings, $p$ is the number of patches, and $d$ refers to the embedding dimension of the patches.

$$X_{pC}^d = \text{Patching}(x_{L,C}) + E_{dp}$$

$$X_{pC}'^d = \text{LayerNorm}(X_{pC}^d + \text{MSA}(X_{pC}^d, X_{pC}^d, X_{pC}^d))$$

$$X''^{d}_{pC} = \text{LayerNorm}(X'^{d}_{pC} + \text{MLP}(X'^{d}_{pC}))$$

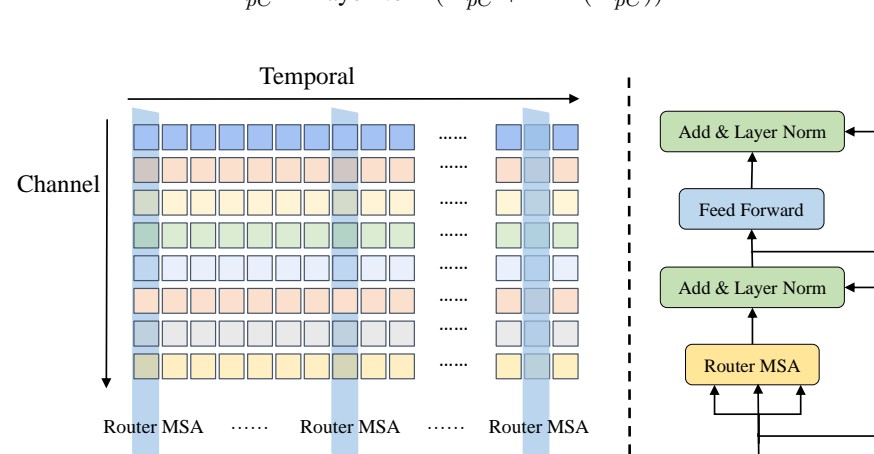

Figure 3: The architecture of the channel module. The left side presents a schematic diagram of the computational logic applied to the data, while the right side illustrates the module structure.

In the cross stage, two cross-channel attention modules (channel modules) are added to the model, while the temporal module and its parameters trained in the temporal stage are retained. The operations of the channel module are shown in Figure2, with the main difference from the temporal module being the replacement of MSA with router MSA. Router MSA is an algorithm proposed by Crossformer that reduces the complexity of cross-channel attention calculations to linear. It uses a set of learnable parameters $r^{d}_{z}$, where $z$ is a predefined constant, as routers to first aggregate information from all channels. Then, each channel computes attention with $r^{d}_{z}$.

$$R = \text{MSA}(r^{d}_{z}, X^{d}_{pC}, X^{d}_{pC})$$

$$X'^{d}_{pC} = \text{MSA}(X^{d}_{pC}, R, R)$$

### 3.3 PARAMETER SHARING OR NOT SHARING

In studies like PatchTST, each dimension of the time series data shares the same model parameters, which is reasonable for most datasets since the dimensions can be regarded as homogeneous. However, this assumption is not appropriate for the weather dataset, as the data includes different types such as temperature and wind speed, which are not homogeneous. Therefore, in our model, we modify the positional encoding for this dataset, replacing $E_{dp}$ with $E^{C}_{dp}$, which means that different channels have distinct positional encodings. This approach allows us to map different types of data into the same vector space.

## 4 EXPERIMENT

### 4.1 DATA

We use seven widely used datasets: Weather, Traffic, Electricity, and ETT datasets (ETTh1, ETTh2, ETTm1, ETTm2), to evaluate the performance of our model. For information on the datasets and how to obtain them, please refer to Autoformer (Wu et al. (2021)).

### 4.2 EXPERIMENT SETTINGS

We selected several of the currently best-performing models as baselines, including TIME-LLM (Jin et al. (2023)), PatchTST (Nie et al. (2022)), DLinear Zeng et al. (2023), FEDformer (Zhou et al. (2022)), Autoformer (Wu et al. (2021)), Informer (Zhou et al. (2021)), and Pyraformer (**?**). We followed the experimental setup used in PatchTST, maintaining the same proportions for the training, validation, and test sets. We chose Mean Squared Error (MSE) as the loss function.

| Models | | DIMS | | TIME-LLM | | PatchTST | | DLinear | | FEDformer | | Autoformer | | Informer | | Pyraformer | |
|---|---|---|---|---|---|---|---|---|---|---|---|---|---|---|---|---|---|
| Metric | | MSE | MAE | MSE | MAE | MSE | MAE | MSE | MAE | MSE | MAE | MSE | MAE | MSE | MAE | MSE | MAE |
| Weather | 96 | **0.144** | **0.196** | 0.147 | 0.201 | 0.149 | 0.198 | 0.176 | 0.237 | 0.238 | 0.314 | 0.249 | 0.329 | 0.354 | 0.405 | 0.896 | 0.556 |
| | 192 | **0.185** | 0.236 | 0.189 | **0.234** | 0.194 | 0.241 | 0.220 | 0.282 | 0.275 | 0.329 | 0.325 | 0.370 | 0.419 | 0.434 | 0.622 | 0.624 |
| | 336 | **0.241** | 0.281 | 0.262 | **0.279** | 0.245 | 0.282 | 0.265 | 0.319 | 0.339 | 0.377 | 0.351 | 0.391 | 0.583 | 0.543 | 0.739 | 0.753 |
| | 720 | 0.306 | 0.327 | **0.304** | **0.316** | 0.314 | 0.334 | 0.323 | 0.362 | 0.389 | 0.409 | 0.415 | 0.426 | 0.916 | 0.705 | 1.004 | 0.934 |
| Traffic | 96 | **0.354** | **0.251** | 0.362 | 0.248 | 0.360 | 0.249 | 0.410 | 0.282 | 0.576 | 0.359 | 0.597 | 0.371 | 0.733 | 0.410 | 2.085 | 0.468 |
| | 192 | **0.372** | **0.255** | 0.374 | **0.247** | 0.379 | 0.256 | 0.423 | 0.287 | 0.610 | 0.380 | 0.607 | 0.382 | 0.777 | 0.435 | 0.867 | 0.467 |
| | 336 | **0.385** | **0.266** | **0.385** | 0.271 | 0.392 | **0.264** | 0.436 | 0.296 | 0.608 | 0.375 | 0.623 | 0.387 | 0.776 | 0.434 | 0.869 | 0.469 |
| | 720 | **0.415** | **0.283** | 0.430 | 0.288 | 0.432 | 0.286 | 0.466 | 0.315 | 0.621 | 0.375 | 0.639 | 0.395 | 0.827 | 0.466 | 0.881 | 0.473 |
| Electricity | 96 | **0.126** | **0.222** | 0.131 | 0.224 | 0.129 | **0.222** | 0.140 | 0.237 | 0.186 | 0.302 | 0.196 | 0.313 | 0.304 | 0.393 | 0.386 | 0.449 |
| | 192 | **0.145** | **0.240** | 0.152 | 0.241 | 0.147 | **0.240** | 0.153 | 0.249 | 0.197 | 0.311 | 0.211 | 0.324 | 0.327 | 0.417 | 0.386 | 0.443 |
| | 336 | **0.152** | 0.250 | 0.160 | **0.248** | 0.163 | 0.259 | 0.169 | 0.267 | 0.213 | 0.328 | 0.214 | 0.327 | 0.333 | 0.422 | 0.378 | 0.443 |
| | 720 | **0.176** | **0.272** | 0.192 | 0.298 | 0.197 | 0.290 | 0.203 | 0.301 | 0.233 | 0.344 | 0.236 | 0.342 | 0.351 | 0.427 | 0.376 | 0.445 |
| ETTh1 | 96 | **0.351** | **0.390** | 0.362 | 0.392 | 0.370 | 0.400 | 0.375 | 0.399 | 0.376 | 0.415 | 0.435 | 0.446 | 0.941 | 0.769 | 0.664 | 0.612 |
| | 192 | **0.390** | **0.414** | 0.398 | 0.418 | 0.413 | 0.429 | 0.405 | 0.416 | 0.423 | 0.446 | 0.456 | 0.457 | 1.007 | 0.786 | 0.790 | 0.681 |
| | 336 | **0.398** | **0.419** | 0.430 | 0.427 | 0.422 | 0.440 | 0.439 | 0.443 | 0.444 | 0.462 | 0.486 | 0.487 | 1.038 | 0.784 | 0.891 | 0.738 |
| | 720 | 0.448 | **0.451** | **0.442** | 0.457 | 0.447 | 0.468 | 0.472 | 0.490 | 0.469 | 0.492 | 0.515 | 0.517 | 0.963 | 0.857 | 0.963 | 0.782 |
| ETTh2 | 96 | **0.265** | 0.330 | 0.268 | **0.328** | 0.274 | 0.337 | 0.289 | 0.353 | 0.332 | 0.374 | 0.332 | 0.368 | 1.549 | 0.952 | 0.645 | 0.597 |
| | 192 | **0.329** | **0.372** | **0.329** | 0.375 | 0.341 | 0.382 | 0.383 | 0.418 | 0.407 | 0.446 | 0.426 | 0.434 | 3.792 | 1.542 | 0.788 | 0.683 |
| | 336 | **0.324** | **0.379** | 0.368 | 0.409 | 0.329 | 0.384 | 0.448 | 0.465 | 0.400 | 0.447 | 0.477 | 0.479 | 4.215 | 1.642 | 0.907 | 0.747 |
| | 720 | **0.372** | **0.413** | **0.372** | 0.420 | 0.379 | 0.422 | 0.605 | 0.551 | 0.412 | 0.469 | 0.453 | 0.490 | 3.656 | 1.619 | 0.963 | 0.783 |
| ETTm1 | 96 | 0.285 | 0.341 | **0.272** | **0.334** | 0.293 | 0.346 | 0.299 | 0.343 | 0.326 | 0.390 | 0.510 | 0.492 | 0.626 | 0.560 | 0.543 | 0.510 |
| | 192 | 0.329 | 0.371 | **0.310** | **0.358** | 0.333 | 0.370 | 0.335 | 0.365 | 0.365 | 0.415 | 0.514 | 0.495 | 0.725 | 0.619 | 0.557 | 0.537 |
| | 336 | 0.360 | 0.393 | **0.352** | **0.384** | 0.369 | 0.392 | 0.369 | 0.386 | 0.392 | 0.425 | 0.510 | 0.492 | 1.005 | 0.741 | 0.754 | 0.655 |
| | 720 | 0.440 | 0.0.436 | **0.383** | **0.411** | 0.416 | 0.420 | 0.425 | 0.421 | 0.446 | 0.458 | 0.527 | 0.493 | 1.133 | 0.845 | 0.908 | 0.724 |
| ETTm2 | 96 | 0.162 | **0.250** | **0.161** | 0.253 | 0.166 | 0.256 | 0.167 | 0.260 | 0.180 | 0.271 | 0.205 | 0.293 | 0.355 | 0.462 | 0.435 | 0.507 |
| | 192 | **0.215** | **0.286** | 0.219 | 0.293 | 0.223 | 0.296 | 0.224 | 0.303 | 0.252 | 0.318 | 0.278 | 0.336 | 0.595 | 0.586 | 0.730 | 0.673 |
| | 336 | **0.267** | **0.321** | 0.271 | 0.329 | 0.274 | 0.329 | 0.281 | 0.342 | 0.324 | 0.364 | 0.343 | 0.379 | 1.270 | 0.871 | 1.201 | 0.845 |
| | 720 | **0.349** | **0.379** | 0.352 | **0.379** | 0.362 | 0.385 | 0.397 | 0.421 | 0.410 | 0.420 | 0.414 | 0.419 | 3.001 | 1.267 | 3.625 | 1.451 |
| 1st count | | **38** | | 22 | | 2 | | 0 | | 0 | | 0 | | 0 | | 0 | |

Table 1: Multivariate long-term forecasting main results with DIMS. The prediction lengths $T \in \{96, 192, 336, 720\}$ . The best results are in **bold**.

For the ETT datasets, the model's input length is 624. For the Traffic and Electricity datasets, the input length is 524, and for the Weather dataset, the input length is 648. On all datasets, the model's prediction lengths are set to 96, 192, 336, and 720. The model uses Revin (Kim et al. (2021)) for normalization to mitigate distribution shifts. For the ETT datasets, the embedding dimension of the seasonal component model is 32, with a hidden layer dimension of 64 for the feedforward neural network. The embedding dimension for the trend component model is 4, with a hidden layer dimension of 8. For the other datasets, the embedding dimension of the seasonal component model is 128, with a hidden layer dimension of 256 for the feedforward neural network. The embedding dimension for the trend component model is 32, with a hidden layer dimension of 64.

## 4.3 MAIN RESULTS

The experimental results across the seven datasets are shown in Table 1. From the results, it is evident that DIMS outperforms other transformer-based models, particularly demonstrating significant improvements on the ETTh1 and Traffic datasets.DIMS achieved first place in 38 metrics, surpassing the 22 metrics of the time-series large language model Time-LLM and the 2 metrics of the PatchTST model.

## 4.4 ABLATION STUDY

We separately removed the seasonal-trend decomposition and channel dependency components to observe the differences in model prediction performance, with the results shown in Table 2. These results indicate that both the seasonal-trend decomposition and channel dependency components positively impact model predictions.

Additionally, for the Weather dataset, we conducted ablation experiments regarding the positional encoding embedding methods, with results shown in Table 3. The results demonstrate that non-shared positional encodings perform better for the Weather dataset.

Regarding the timing of the inclusion of the channel module, we also performed ablation experiments, and the results are presented in Table 4. The experimental data indicate that delaying the addition of the channel module can lead to optimal prediction performance.

| Variants | DIMS | | DIMS w/o Cross | | DIMS w/o Decomp. | |
|---|---|---|---|---|---|---|
| Metric | MSE | MAE | MSE | MAE | MSE | MAE |
| ETTh1 96 | **0.351** | **0.390** | **0.351** | **0.390** | 0.356 | 0.393 |
| ETTh1 192 | **0.390** | **0.414** | 0.392 | 0.415 | 0.413 | 0.437 |
| ETTh1 336 | **0.398** | **0.419** | 0.400 | 0.420 | 0.415 | 0.443 |
| ETTh1 720 | 0.448 | **0.451** | **0.440** | **0.451** | 0.459 | 0.478 |
| Traffic 96 | 0.354 | 0.251 | 0.358 | 0.244 | 0.351 | **0.243** |
| Traffic 192 | **0.372** | 0.255 | 0.373 | 0.255 | 0.374 | **0.252** |
| Traffic 336 | **0.385** | 0.266 | 0.390 | 0.264 | 0.390 | **0.261** |
| Traffic 720 | **0.415** | 0.283 | 0.423 | **0.283** | 0.430 | 0.285 |
| Electricity 96 | **0.126** | **0.222** | 0.129 | 0.223 | 0.129 | 0.226 |
| Electricity 192 | 0.145 | 0.240 | 0.148 | 0.241 | **0.144** | **0.237** |
| Electricity 336 | **0.152** | **0.250** | 0.161 | 0.257 | 0.161 | 0.256 |
| Electricity 720 | **0.176** | **0.272** | 0.204 | 0.300 | 0.196 | 0.289 |
| 1st count | **17** | | 5 | | 6 | |

Table 2: Ablation experiment results of the DIMS model. ”w/o Cross” indicates the removal of the channel module, and ”w/o Decomp.” indicates the removal of the seasonal-trend decomposition. The best results are in **bold**.

| Variants | DIMS | | DIMS w/o Cross | | DIMS w/o Decomp | | DIMS share pos. | |
|---|---|---|---|---|---|---|---|---|
| Metric | MSE | MAE | MSE | MAE | MSE | MAE | MSE | MAE |
| Weather 96 | 0.144 | 0.196 | **0.142** | **0.194** | 0.143 | 0.194 | 0.147 | 0.199 |
| Weather 192 | **0.185** | **0.236** | **0.185** | **0.236** | 0.190 | 0.240 | 0.189 | 0.238 |
| Weather 336 | **0.241** | 0.281 | 0.242 | 0.281 | **0.241** | 0.281 | 0.241 | **0.280** |
| Weather 720 | **0.306** | **0.327** | 0.310 | 0.330 | 0.319 | 0.337 | 0.315 | 0.334 |

Table 3: Ablation experiment results of the DIMS model on the weather dataset. ”w/o Cross” indicates the removal of the channel module, ”w/o Decomp.” indicates the removal of the seasonal-trend decomposition, and ”share pos.” indicates the use of shared positional encoding across channels. The best results are in **bold**.

| Variants | Delay Cross-Stage | Sync. Cross-Stage | w/o Cross-Stage |
|---|---|---|---|
| Metric | MSE | MSE | MSE |
| Traffic 96 | **0.303** | 0.309 | 0.310 |
| Traffic 192 | **0.328** | **0.328** | 0.335 |
| Traffic 336 | **0.349** | 0.351 | 0.353 |
| Traffic 720 | **0.381** | 0.392 | 0.392 |
| Electricity 96 | **0.107** | 0.108 | 0.108 |
| Electricity 192 | **0.113** | 0.116 | 0.115 |
| Electricity 336 | **0.121** | 0.123 | 0.123 |
| Electricity 720 | 0.135 | **0.134** | 0.138 |

Table 4: Comparison experiment results of the timing for adding the channel module. ”Delay Cross-Stage” indicates that the Cross-Stage is conducted later, ”Sync. Cross-Stage” indicates that the Cross-Stage is synchronized with the channel-independent temporal stage, and ”w/o Cross-Stage” indicates the removal of the channel module. The best results are in **bold**.

## 5 CONCLUSION

In this study, we explored the key factors contributing to the overfitting of time series prediction models and proposed two novel and straightforward improvements: seasonal-trend independence and channel dependence. We conducted detailed experiments on the proposed methods, and the results confirm that our approach surpasses the prediction performance of existing state-of-the-art transformer-based models. We also performed multiple ablation experiments to validate the effectiveness of each proposed method.

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
