# OpenReview forum: "DIMS: Channel-Dependent and Seasonal-Trend Independent Transformer Using Multi-Stage Training for Time Series Forecasting"
_ICLR.cc/2025/Conference — ICLR 2025 Conference Withdrawn Submission_

### Official Review · Reviewer_uixy · 2024-10-23

**Soundness:** 2
**Presentation:** 2
**Contribution:** 2
**Rating:** 3
**Confidence:** 5

**Summary:**

This paper introduces a new multivariate time series forecasting method called DIMS, which is a two-stage training paradigm. First, in the first stage, channel independent modeling is used, while seasonal-trend decomposition is optimized independently. In the second stage, a channel dependency modeling module is added to capture the correlation between channels.

**Strengths:**

1. The method proposed in this paper is very simple and easy to understand.

**Weaknesses:**

1. The method proposed in this paper is not novel enough, it is a simple recombination of existing works and the improvement part is not innovative enough. All modules or technologies used in this paper are derived from existing work and improved two-stage training methods is not enough innovation and the performance gain is not obvious.
2. There are some problems with writing, there is an error in the literature citation of RLinear on the page 2.
3. This paper simply believes that the existing seasonal-trend decomposition module has shortcomings, and proposes independent optimization operations in the first stage, but this view is not supported by ablation experiments.
4. The current experiments in this paper are seriously insufficient to prove the effectiveness of the proposed method. More experiments need to be added, such as adding the proposed improved seasonal-trend decomposition module based on models such as FEDformer to verify its effectiveness. At the same time, this article should not be limited to verification of effectiveness based on the transformer architecture, and the proposed method should be used as a plug-in for more extensive verification.
5. In both table 2 and table 4, there are ablation experiments to remove the channel module. Why are the experimental results of the traffic and electricity datasets in the two tables different?

**Questions:**

See Weaknesses

---

### Official Review · Reviewer_ugYY · 2024-11-04

**Soundness:** 2
**Presentation:** 2
**Contribution:** 2
**Rating:** 3
**Confidence:** 4

**Summary:**

In this study, the authors propose DIMS, a time series forecasting model that uses multi-stage training to capture inter-channel dependencies while ensuring the independence of seasonal and trend components. Experiments show some improvements compared to the state-of-the-art transformer-based models on several datasets.

**Strengths:**

1.	The author combines the strategies of time decomposition and cross-channel modeling.
2.	The proposed methods are evaluated through ablation studies.
3.	The proposed methods are verified to outperform benchmark methods.

**Weaknesses:**

1. The motivation of this study is not clear. The authors claim existing methods have limitations of over-fitting problem, which is not explained convincingly. The authors didn’t provide insightful data analysis about the current over-fitting problem and how their proposed methods perform. For example, the authors can analyze learning curves to show training vs. validation performance over epochs for baseline models compared to DIMS.

2. The authors combine cross-channel modeling and time decomposition techniques but haven’t compared their proposed methods to corresponding stat-of-the-art benchmarks. For example, the authors largely refer to Crossformer to model cross-channel dependencies, but they haven’t compared DIMS to Crossformer in the experiment section. Similarly, the authors are expected to compare DIMS with other advanced time decomposition methods to demonstrate its strength. I suggest the authors compare their methods with [1][2] and explain why their design is better.

[1] Crossformer: Transformer utilizing cross-dimension dependency for multivariate time series forecasting

[2] Timemixer: Decomposable multiscale mixing for time series forecasting

3. DIMS only achieves trivial improvement in forecasting accuracy compared to the benchmarks, and such improvement may be raveled out if authors carried out repeated experiments (traditionally 3-5 times). The authors should provide sufficient statistical confidence to show the consistent superiority of the proposed methods.

4. It is suggested to supplement the theoretical analysis to further improve the technical contribution of the proposed method. For example, the authors should explain how the proposed decomposition and cross-channel modeling method can alleviate the over-fitting problem by analyzing the generalization bound.

5. Several references are incorrectly cited, such as RLinear in line 94, and Pyraformer in line 268.

**Questions:**

1.	The motivations should be further demonstrated with data analysis.
2.	The proposed methods should be compared to state-of-the-art cross-channel modeling methods and time decomposition methods.
3.	The proposed methods should be analyzed theoretically.
4.	The effectiveness of the proposed methods should be evaluated through repeated experiments.

---

### Official Review · Reviewer_Deza · 2024-11-05

**Soundness:** 2
**Presentation:** 2
**Contribution:** 3
**Rating:** 3
**Confidence:** 3

**Summary:**

In this paper, authors raise a time series forecasting technique, DIMS, which can absorb the advantages of channel dependence and channel independence at the same time, with the help of seasonal-trend decompositions. Experiments show that this method outperform some state of art baseline methods in various dataset.

**Strengths:**

1. Although the concept of channel independence/dependence and decomposition techniques are not very innovative, it is a novel combination to capture inter-channel dependencies while ensuring the independence of seasonal and trend components.
2. Figure 1 and figure 2 clearly explain the key methodology of DIMS.
3. A nice discussion about the related work section to explain the development of transformer in time series forecasting.

**Weaknesses:**

1. This paper needs proofread. (e.g., line 094, page 2).
2. For experiments section, it misses an important baseline methods, itransformer since it is really related to channel independence. Additionally, authors also discuss many non-transfomer methods for TSF and they might be also used as baselines, such as FiTS (Xu et al. (2023)).

Liu, Yong, Tengge Hu, Haoran Zhang, Haixu Wu, Shiyu Wang, Lintao Ma, and Mingsheng Long. "itransformer: Inverted transformers are effective for time series forecasting." arXiv preprint arXiv:2310.06625 (2023).


3. For the ablation study section, it is better to do more on all the datasets. For example, in table 3, authors only use weather data. It would be better to conclude more dataset to suppport their argument.
3. One key contribution, as mentioned in abstract, is to overcome the overfitting problems in transformer models. However, in experiments, authors do not explain or experiment how DIMS solve the overfitting problem. It is better to, for example, provide some training errors, validation errors and testing errors of different transformer models to prove that this problem mitigate the overfitting problem.

**Questions:**

1. See Weaknesses.
2. For decomposition, what is the difference between DIMS's approach and Dlinear's approach?
3. Are there any experimental result for 3.3 section, parameter sharing and/or not sharing?
4. Why you use Time-LLM as one of your baseline models? Could you speculate why TIme-LLM outperforms DIMS a lot in ETTm1?

---

### Official Review · Reviewer_qedi · 2024-11-08

**Soundness:** 2
**Presentation:** 1
**Contribution:** 1
**Rating:** 1
**Confidence:** 4

**Summary:**

This paper presents a time series forecasting model that leverages multi-stage training to capture inter-channel dependencies while ensuring the independence of seasonal and trend components. The experiment results showed the effectiveness of the proposed method.

**Strengths:**

* Time series forecasting is an important problem to investigate.
* The idea of capturing the inter-channel dependency and ensuring the independence of the seasonal and trend components seems interesting.

**Weaknesses:**

* The writing of this paper needs improvement. Many parts are not articulated (see below questions) clearly and several citations are problematic, e.g., RLinear [?] and Pyramformer [?].
* Several related works are not mentioned or compared (see below questions).
* The key idea in this paper is based on the combination of seasonal-trend decomposition and cross-channel relationship in crossformer. However, both ideas have been used in previous works. Therefore, the overall technical novelty is limited.

**Questions:**

1. The details of the decomposition are not clearly articulated. How did you perform the seasonal-trend decomposition? What are your parameters for the moving average? Is it conducted over the entire training/vali/test dataset first, then model training is performed? Or it is conducted over the input window?

2. Some recent works should be mentioned or compared e.g.,

[1] Unified Training of Universal Time Series Forecasting Transformers, ICML 2024

[2] Chronos: Learning the Language of Time Series. arXiv preprint arXiv:2403.07815.

[3] TimeMixer: Decomposable Multiscale Mixing for Time Series Forecasting, ICLR 2024

3. The authors claim that decomposition can help resolve the overfiting. However, there is no clear evidence to support this claim.

4. Although the ablation study has been provided, the results do not show significant differences between different variants.

---

### Note · Authors · 2024-11-18

I have read and agree with the venue's withdrawal policy on behalf of myself and my co-authors.